# GENERATIVE LATENT VIDEO COMPRESSION

## ABSTRACT

Perceptual optimization is widely recognized as essential for neural compression, yet balancing the rate–distortion–perception tradeoff remains challenging. This difficulty is especially pronounced in video compression, where frame-wise quality fluctuations often cause perceptually optimized neural video codecs to suffer from flickering artifacts. In this paper, inspired by the success of latent generative models, we present **G**enerative **L**atent **V**ideo **C**ompression (**GLVC**), an effective framework for perceptual video compression. GLVC employs a pretrained continuous tokenizer to project video frames into a perceptually aligned latent space, thereby offloading perceptual constraints from the rate–distortion optimization. We redesign the codec architecture explicitly for the latent domain, drawing on extensive insights from prior neural video codecs, and further equip it with innovations such as unified intra/inter coding and a recurrent memory mechanism. Experimental results across multiple benchmarks show that GLVC achieves state-of-the-art performance in terms of DISTS and LPIPS metrics. Notably, our user study confirms GLVC rivals the latest neural video codecs at nearly half their rate while maintaining stable temporal coherence, marking a step toward practical perceptual video compression.

## 1 INTRODUCTION

The primary goal of video compression is to achieve an optimal tradeoff between bitrate and visual quality, thereby significantly reducing the cost of video transmission and storage. Advances in deep learning have enabled neural video compression (NVC) methods (Li et al., 2023; 2024b) to achieve rate-distortion performance comparable to, or even surpassing, traditional compression standards such as H.266 (Bross et al., 2021) and ECM (JVET, 2025), the prototype of the next-generation standard. Recent work (Jia et al., 2025) further demonstrates the potential of NVC for practical deployment, highlighting its promise for high-efficiency video coding in real-world applications.

Despite this remarkable progress, NVCs remain far from reaching their full potential. One promising direction explored in recent works (Mentzer et al., 2020; 2022a) is perceptually optimized compression, which is motivated by the fact that commonly used objective metrics such as peak signal-to-noise ratio (PSNR) do not fully align with human perception (Wang et al., 2004). While perceptual optimization has led to notable advances in neural image compression, both theoretically (Blau & Michaeli, 2019) and in implementations (Muckley et al., 2023), extending these gains to video compression remains challenging. Existing perceptual video compression methods often exhibit severe temporal inconsistencies in the reconstructed details, resulting in flickering artifacts that degrade perceptual quality, as reported in prior literature (Qi et al., 2025).

Such flickering artifacts become even more pronounced at extremely low bitrates. This is perhaps unsurprising, as most perceptual-optimized neural video codecs rely on video generative adversarial networks (GANs) (Goodfellow et al., 2014), whose training is often unstable (Mescheder et al., 2018). Compared to image compression, applying GANs to video compression introduces additional challenges: the model must generate temporally coherent frames while simultaneously balancing perception and rate-distortion. Keeping such a balance in videos is especially difficult because rate-distortion characteristics vary inherently across frames, primarily due to error propagation, which causes quality degradation in later compressed frames (Li et al., 2024b). Moreover, the perceptual details synthesized by GANs must be preserved consistently over time. Otherwise, flickering artifacts are easily produced due to a failure to maintain coherent inter-frame dependencies.

Figure 1: Example of decoded videos. Click to play the video. Best viewed with Adobe Reader.

To address these challenges, inspired by the success of latent generative models (Rombach et al., 2022; Brooks et al., 2024), we establish the framework of **G**enerative **L**atent **V**ideo **C**ompression (**GLVC**). We employ a tokenizer to first map raw video frames from pixel space into a continuous latent space, where perceptual detail synthesis and temporal smoothness are learned during pretraining. A separate latent compression module then encodes these latent representations under rate-distortion constraints. Unlike prior generative latent compression approach (Qi et al., 2025) that relies on discrete vector-quantized latents and suffers from severe flickering artifacts, GLVC adopts a continuous latent space that proves crucial for maintaining temporal coherence. By decoupling perceptual detail synthesis (handled by the continuous tokenizer) from bitrate-aware compression (performed by the latent codec), GLVC achieves robust temporal consistency and strong perceptual quality even at very low bitrates.

On top of this framework, we highlight our carefully designed latent codec module. Building on insights from prior neural video codec (Jia et al., 2025), we adapt and redesign key components to meet the demands of latent-domain compression. We further introduce two technical innovations: first, a unified intra-/inter-frame compression model that reduces parameter redundancy; and second, a recurrent memory mechanism that continuously updates and retains semantic latent representations from past frames. This memory provides long-range temporal context, enabling more effective modeling of video history and further suppressing flickering artifacts.

GLVC achieves substantially better perceptual quality than prior NVCs. On the UVG dataset (Mercat et al., 2020), it delivers 94.7% BD-rate savings (Bjontegaard, 2001) compared with the latest NVC (Jia et al., 2025) when evaluated by DISTS (Ding et al., 2020). While we acknowledge that such dramatic gains partly reflect mismatches between automated perceptual metrics and human judgments, we validate the results through a user study. The study shows that GLVC outperforms both traditional and other neural codecs at similar bitrates, and even rivals the latest NVCs while operating at nearly half their rate. We hope this latent compression framework marks a significant step toward practical perceptual video compression.

In summary, our contributions are:

- We establish a generative latent video compression framework with a continuous tokenizer, preserving temporal consistency and perceptual quality even at very low bitrates.

- We redesign the codec for latent-domain compression, introducing innovations such as unified intra/inter coding and a recurrent memory mechanism for long-range consistency.

- We demonstrate the state-of-the-art perceptual video compression performance, both quantitatively and quatitatively validated by user studies.

## 2 BACKGROUND AND MOTIVATION

### 2.1 PERCEPTUAL QUALITY

Perceptual quality has been widely studied across various domains (Blau & Michaeli, 2018; Fang et al., 2020), and is often referred to by alternative names such as *realism* in generative modeling (Fan et al., 2017; Theis, 2024). It is defined in Blau & Michaeli (2018) as the extent of an output sample $\hat{x}$ to which humans perceive it as realistic or natural, regardless of its similarity to the input $x$. This aligns with the class of no-reference quality assessment methods (Mittal et al., 2012a;b).

A common theoretical formulation expresses perceptual quality as the divergence between the distribution of real data $p_X$ and that of generated or reconstructed data $p_{\hat{X}}$:

$$d(p_X, p_{\hat{X}}), \tag{1}$$

where $d(\cdot, \cdot)$ can be any distributional distance metric such as Kullback–Leibler divergence or Wasserstein distance. In practice, GANs (Goodfellow et al., 2014) are frequently used to minimize such divergences, yielding high perceptual quality in synthesis and compression tasks.

However, because perceptual quality is inherently no-reference, it does not always correlate with full-reference distortion metrics such as PSNR or MS-SSIM (Wang et al., 2004). Minimizing distortion may degrade perceptual quality, and vice versa. This tension is known as the *perception-distortion tradeoff* (Blau & Michaeli, 2018), and it becomes even more complex in the context of lossy compression, where another term *rate* must also be considered.

### 2.2 THE RATE–DISTORTION–PERCEPTION TRADEOFF

For lossy compression, the classical rate-distortion function (Cover, 1999) is given by:

$$R(D) = \min I(X; \hat{X}) \quad \text{s.t.} \quad \mathbb{E}[\Delta(X, \hat{X})] \leq D, \tag{2}$$

where $I(X; \hat{X})$ denotes the mutual information between source and reconstruction, and $\Delta$ is a distortion measure. Blau & Michaeli (2019) extended this framework to incorporate perceptual constraints, resulting in the rate-distortion-perception (R-D-P) function:

$$R(D, P) = \min I(X; \hat{X}) \quad \text{s.t.} \quad \mathbb{E}[\Delta(X, \hat{X})] \leq D, \quad d(p_X, p_{\hat{X}}) \leq P, \tag{3}$$

where $P$ constrains the perceptual distance between the real and reconstructed distributions. In practice, it is intuitive to optimize a combined loss that balances rate, distortion and perception:

$$\mathcal{L} = R(y) + \alpha \cdot \Delta(X, \hat{X}|y) + \beta \cdot d(p_X, p_{\hat{X}|y}), \tag{4}$$

where $R(y)$ is the estimated bitrate and $\alpha$, $\beta$ are tradeoff weights. However, although Lagrangian methods work well for rate-distortion optimization (Eq. 2), incorporating the perception term introduces significant challenges especially for video. Usually, earlier compressed frames may exhibit high quality, while later ones degrade due to error propagation. It means the Lagrange weights ($\alpha, \beta$) require complex heuristic settings across different frames (Li et al., 2024b). Note some works (Yan et al., 2021; Zhang et al., 2021) propose universal representations to facilitate R-D-P optimization, but extending them to video remains hard when considering perception loss functions with joint distributions of all video frames (Salehkalaibar et al., 2023). A more structured approach is needed to handle temporal quality variation and preserve stable perceptual quality over time.

### 2.3 DECOUPLING SEMANTIC COMPRESSION FROM PERCEPTION

The success of visual generative models for both images (Esser et al., 2021; Rombach et al., 2022) and videos (Brooks et al., 2024; Yang et al., 2024; Kong et al., 2024) has largely relied on the latent generation framework. As observed in (Rombach et al., 2022), visual information can be roughly decomposed into two components: semantic information, which governs appearance or structure, and is closely tied to MSE/PSNR distortion, and perceptual information, which governs realism and is more loosely related to distortion. This behavior is evident in image diffusion models (Ho et al., 2020) (see Fig. 2a): they first generate semantic structure in the low-rate regime, consuming relatively few bits, and then progressively synthesize perceptual details in the high-rate regime,

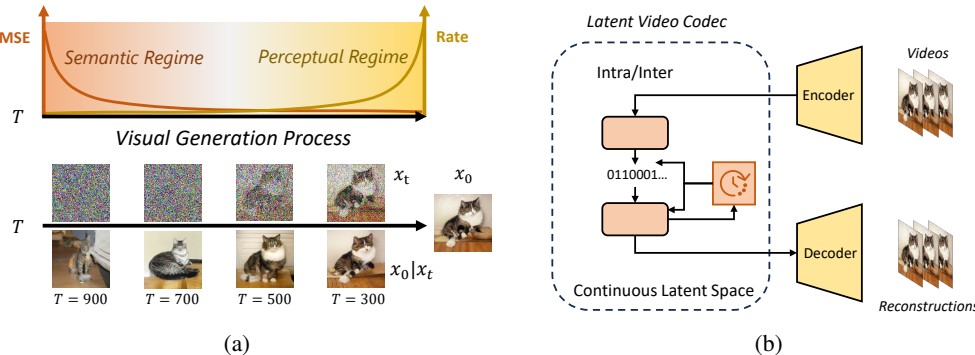

(a)                                    (b)

Figure 2: (a) Visual generation process illustrated by diffusion models (Ho et al., 2020). Information emerges first in the semantic regime, where bits allocated are strongly tied to distortion (MSE/PSNR). In the subsequent perceptual regime, a larger share of bits is consumed to enrich realism but contributing less to distortion. (b) GLVC performs rate–distortion optimization in a continuous semantic latent space while offloading perceptual detail synthesis to the tokenizer

which consumes higher rate. It is also verified in (Theis et al., 2022) that compressing semantic information consumes much fewer bits than compressing perceptual information.

Extending this perspective to videos, motion information should likewise be regarded as semantic, since distortion metrics such as PSNR are highly sensitive to motion fidelity. This motivates us to establish a latent video compression framework that separates semantic optimization from perceptual constraints. In particular, bits allocated in the semantic regime directly govern fidelity, while bits consumed in the perceptual regime are substantial but mainly contribute to enhancing realism. We therefore place rate–distortion optimization within the semantic regime to compress appearance and motion, and offload perceptual quality to a separate tokenizer, which enables more stable optimization and preserve temporal coherence.

## 3 GENERATIVE LATENT VIDEO COMPRESSION

### 3.1 FRAMEWORK

Building on these observations, we introduce Generative Latent Video Compression (GLVC), a framework that decouples rate–distortion optimization of video semantics from perceptual learning. As illustrated in Fig. 2b, the system consists of two stages. First, a pretrained tokenizer maps raw video frames into a continuous latent space, serving as the perceptual front-end that synthesizes realistic details in the realism regime. Second, a latent video compression module encodes these latent representations, targeting semantic fidelity and rate efficiency in the semantics regime.

Unlike prior works (Jia et al., 2024; Qi et al., 2025) that map images/videos into vector-quantized (VQ) latent spaces, we find that a continuous latent space is crucial for maintaining temporal consistency (as shown later in Sec. 5.3). Empirically, this is because VQ tokenizers discretize latent representations in a non-smooth manner, which breaks temporal continuity and intensify flickering artifacts as in Qi et al. (2025). In this work, we adopt the continuous Wan tokenizer (Team et al., 2025) as the perceptual front-end. The tokenizer is pretrained with an adversarial objective to ensure realistic reconstructions, and it maps raw video frames of size $3 \times T \times H \times W$ (with $T = 4K + 1$, $K \in \mathbb{N}$) into latent representations of size $16 \times (K + 1) \times H/8 \times W/8$. In the following subsection, we describe our engineered latent video compression module, which performs effective rate–distortion optimization on these continuous latent representations.

### 3.2 LATENT VIDEO COMPRESSION MODULE

Once video frames are transformed into the semantic latent space, the next challenge is to design a compression module that effectively exploits both spatial and temporal redundancies of latent representations. To this end, we take experience from the DCVC series Li et al. (2021; 2023; 2024b) and recent advances in neural video compression (Jia et al., 2025), building the latent video codec. Our

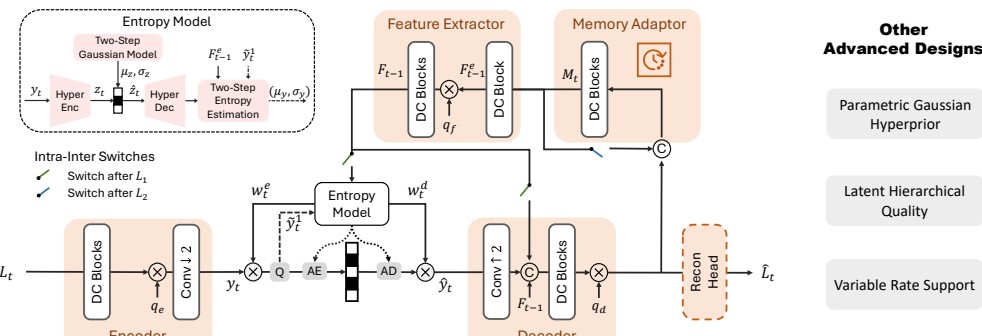

Figure 3: Our designed latent video compression module. It unifies both intra and inter compression and incorporates a recurrent memory mechanism that stores the historical semantic information from previously compressed latent representations.

latent codec integrates intra- and inter-frame compression into a unified architecture and introduces a recurrent memory mechanism to maintain long-range temporal coherence.

**Unified Intra- and Inter- Latent Codec**  Unlike previous DCVC series that employ separate networks or modules for intra and inter coding, we adopt a unified architecture that compress video latent representations within a single framework. Most modules are shared across all latents, with only minimal switchable components to accommodate differences between the first (intra) latent and subsequent (inter) latents. For instance, when compressing the first latent without any temporal context, the model activates intra-specific convolution layers. For later latents, when context from previously compressed latents is available, the model switches to inter-convolution layers that process concatenated features from current and past latents, as illustrated in Fig. 3.

Joint training of intra- and inter-prediction enables shared parameters to benefit from both spatial and temporal supervision, fostering better feature alignment. This unified design not only reduces model complexity, but also supports the memory mechanism introduced later, as all latents are encoded within a consistent latent space.

**Recurrent Memory Mechanism**  To further enhance temporal modeling, we introduce a recurrent memory mechanism that dynamically store the historical semantic information from the previously decoded latent representations. The memory learned by the memory adaptor in Fig. 3 acts as a persistent context buffer, enabling the model to access long-range semantic information. At each time step, the memory is recurrently updated with the feature before the final reconstruction head, and selectively read to guide the inter prediction. Different from the temporal context (Li et al., 2021) that is derived only from the previous frame the memory here is recurrently updated and buffered, which could recept to a long history information ideally.

This mechanism is designed to memorize the semantic correlation in continuous latent space. Unlike the video states in prior work (Rippel et al., 2019) that are concatenated into encoder, the memory information here facilitates entropy coding and better decoding. To coordinate with the unified intra- and inter-latent compression module, we also add a switch in the memory adaptor. When compressing the second latent $L_2$, it directly takes the feature before final reconstruction from the first latent $\hat{L}_1$ into the memory adaptor. Since the third latent, the memory is updated with input as the concatenation of the previous memory and the current decoded feature.

**Other Advanced Designs**  We further integrate several complementary techniques to enhance compression efficiency and robustness, particularly in extremely low-bitrate regimes. First, we observe that the factorized hyperprior can dominate the overall bitrate when operating at very low rates. To mitigate this, we replace it with a parametric Gaussian model, analogous to the parametric motion compression adopted in (Guo et al., 2023), thereby reducing hyperprior overhead. Second, we adopt a latent hierarchical quality strategy (Li et al., 2024b), which dynamically adjusts the rate–distortion balance during training. This helps alleviate error propagation and maintain quality across long sequences. Third, following (Li et al., 2024b; Jia et al., 2024), we incorporate variable bitrate control via a learnable quantization matrix that is jointly shared by the decoder and entropy model, ensuring flexible and practical deployment.

In summary, the proposed GLVC provides an effective solution for perceptual video compression. The latent video compression framework, together with the carefully engineered latent video codec, enables rate–distortion optimization in the semantic regime while offloading perceptual realism to the tokenizer. This design preserves strong temporal consistency, further enhanced with innovations of unified intra–inter latent compression and the recurrent semantic memorization mechanism. Collectively, these innovations allow GLVC to achieve encouraging results for perceptual-optimized neural video compression, as validated by extensive experimental results in Sec. 5.

## 3.3 TRAINING STRATEGY

Since we leverage a pretrained video tokenizer (Team et al., 2025) to map video frames into semantic latent space, we only need to train the latent video codec. Our training generally consists of a rate-distortion training stage and a short finetuning stage. The latent video codec is first optimized for the tradeoff between latent distortion and bitrate:

$$\mathcal{L}_{\text{latent}} = \lambda_{\text{rate}} \cdot \mathcal{R}(y) + \Delta(L_{1:t}, \hat{L}_{1:t}|y), \tag{5}$$

where $\mathcal{R}(y)$ is the rate, and $\lambda_{\text{rate}}$ is hyperparameters to control the tradeoff between rate and distortion. In theory, given that the continuous tokenizer outputs Gaussian-distributed latents $L$, one could maximize the likelihood of the decoded latents $\hat{L}$ as distortion measures. However, we empirically observed that the variance of $L$ from the video tokenizer is extremely small, making likelihood maximization ineffective. Neither direct use of the original variance nor truncated variance yielded better results. Therefore, we adopt MSE loss to calculate $\Delta(L_{1:t}, \hat{L}_{1:t}|y)$ in practice, which is found stable and effective for minimizing latent distortion. The settings of latent hierarchical quality are illustrated in Appendix A.1. Other training details are the same to (Jia et al., 2025).

After training with the rate–distortion objective, the latent video codec establishes a fixed compression rate. We then unfreeze only the reconstruction head inside the codec (marked as "Recon Head" in Fig. 3), while keeping all other components frozen (including the tokenizer). Since the bitrate has already been determined by the encoder and in-loop codec, the reconstruction head is finetuned solely to improve perceptual fidelity. Following (Esser et al., 2021), we use the L1 loss, perceptual loss (Zhang et al., 2018) and adversarial loss (Goodfellow et al., 2014) to finetune as

$$\mathcal{L}_{\text{finetune}} = \Delta(X_{1:t}, \hat{X}_{1:t}|y) + \lambda_{\text{perceptual}} \cdot \mathcal{L}_{\text{perceptual}}(L_{1:t}, \hat{L}_{1:t}|y) + \lambda_{\text{adv}} \cdot \mathcal{L}_{\text{adv}}(L_{1:t}, \hat{L}_{1:t}|y). \tag{6}$$

Note we do not incorporate the tokenizer's decoder when conduct finetuning, as we experimentally found it degraded performance, likely due to our small finetuning scale.

In short, the above training strategy first allows the latent video codec to learn stable rate–distortion optimization, and then refines perceptual quality through the reconstruction head, achieving realistic reconstructions while preserving the compression rate.

## 4 RELATED WORK

**Neural Video Compression**  Neural video compression (NVC) has progressed rapidly in recent years. Early neural image compression models established the field's foundation by introducing essential techniques such as quantization (Ballé et al., 2017) and entropy modeling (Minnen et al., 2018; He et al., 2021). These techniques have since been extended to video compression, with successive methods improving rate-distortion performance through advances in motion prediction (Lu et al., 2019; Agustsson et al., 2020; Hu et al., 2021; Guo et al., 2023), latent entropy modeling (Ho et al., 2022; Li et al., 2023; 2024b), and framework design (Mentzer et al., 2022b; Chen et al., 2021). Recent advances such as implicit motion modeling (Jia et al., 2025) demonstrate that NVC can be both efficient and practically deployable. Most existing NVC approaches remain optimized primarily for MSE, leaving perception-aligned compression as a promising and challenging frontier.

**Perceptual Compression**  Perceptual compression can be formulated as an optimization of the rate–distortion–perception tradeoff (Blau & Michaeli, 2019). Theoretical analyses suggest that, with universal representations (Zhang et al., 2021; Salehkalaibar et al., 2023), perfect realism can be achieved at the same rate with only twice the MSE distortion of an MSE-optimized codec (Yan et al., 2021). In practice, perceptual image compression has achieved strong results when coupled

with GANs (Muckley et al., 2023), and more recently with diffusion models (Careil et al., 2024) and large multimodal models (Li et al., 2024a). However, extending these successes to video compression remains challenging and existing approaches often struggle to maintain temporal coherence. Evidence for this gap can be found in the CLIC competition series (Competition, 2025), which has evaluated perceptual video quality via human ratings since 2022. Top entries (Zhao et al., 2024) in this competition still relied on post-processing traditional codecs, because existing perceptual video compression solutions still suffers from issues such as notable flickering artifacts.

**Latent Generative Models**  Early work VQGAN (Esser et al., 2021) pioneered visual tokenization for high-quality image synthesis, laying the groundwork for latent generative modeling. Subsequent latent diffusion models (Rombach et al., 2022) popularized this latent generation framework. In the video domain, large-scale models exemplified by Sora (Brooks et al., 2024) have further demonstrated the potential of latent generative approaches. The success of these models stems from the observation mentioned in Sec. 2.3. In this paper, we extend these insights to compression by establishing a generative latent video compression framework that explicitly decouples semantic fidelity from perceptual realism. Notably, while GLC (Jia et al., 2024) explores discrete VQ tokenizers for perceptual image compression, its extension to video (Qi et al., 2025) suffers from pronounced flickering artifacts, not convincing enough to establish the framework of latent video compression.

## 5 EXPERIMENTS

### 5.1 SETTINGS

**Datasets**  Following prior work in neural video compression (Lu et al., 2019; Li et al., 2021; Jia et al., 2025), we train the latent video codec on the Vimeo dataset (Xue et al., 2019), optimizing it for the latent rate–distortion tradeoff. For the subsequent fine-tuning stage, we employ a medium-scale, high-quality video dataset to adapt the out-loop reconstruction head. Specifically, we use a 1M subset of HD-VILA (Xue et al., 2022), filtered using the approach in Wang et al. (2025). For evaluation, we follow standard NVC benchmarks and report results on JVET Class B, C, D, and E (Flynn et al.), UVG (Mercat et al., 2020), and MCL-JCV (Wang et al., 2016).

**Implementation Details**  To enable variable bitrate control in our perceptual-oriented framework, we randomly assign quantization parameters (QPs) in the range [0, 31] for each training iteration. Training is staged by gradually increasing the number of frames: starting from a single frame and eventually to 129 frames, corresponding to 1 latent feature initially and up to 33 latent features in temporal axis. Since the Wan tokenizer (Team et al., 2025) is pretrained in RGB space, our entire model is trained and evaluated directly on RGB frames for consistency.

**Benchmark Settings**  We compare the proposed GLVC with both the MSE-optimized NVC methods, including DCVC-RT (Jia et al., 2025), DCVC (Li et al., 2021), DVC (Lu et al., 2019), and and perceptual-optimized NVC methods including PLVC (Yang et al., 2020) and GLC-video (Qi et al., 2025). We also built baselines with traditional codcs including HM (HM, 2021), VTM (VTM, 2021) and ECM (JVET, 2025), which represents the best H.265, H.266 performance and the prototye of next-generation traditional codec. We report all results in this paper for the first 96 frames of each video sequence. As objective metrics such as PSNR could not fully reflect the perceptual quality of the generated videos, we report the LPIPS (Zhang et al., 2018) and DISTS scores (Ding et al., 2020) as the main results. In addition, we conduct user study following the settings in (Mentzer et al., 2022a), which are illustrated in details as in Appendix A.2.

### 5.2 RESULTS

**Quantitative Comparisons**  As shown in Figure 4, when evaluated in terms of DISTS and LPIPS, the proposed GLVC significantly outperforms prior MSE-optimized neural video compression methods and perceptual-optimized neural video codecs. Specifically,when measured by DISTS on UVG dataset, GLVC achieves 94.7% BD savings against the latest DCVC-RT (Jia et al., 2025) and 66.5% BD savings compared with the latest perceptually optimized method GLC-video (Qi et al., 2025) (despite strong flickering artifact in it). Other BD rate comparisons and RD curves can be found in Appendix A.3. It is noteworthy that our GLVC could achieve comparable PSNR value at low bitrate, which demonstrates it well preserves semantic fidelity. Despite the excellent quantitative results, it

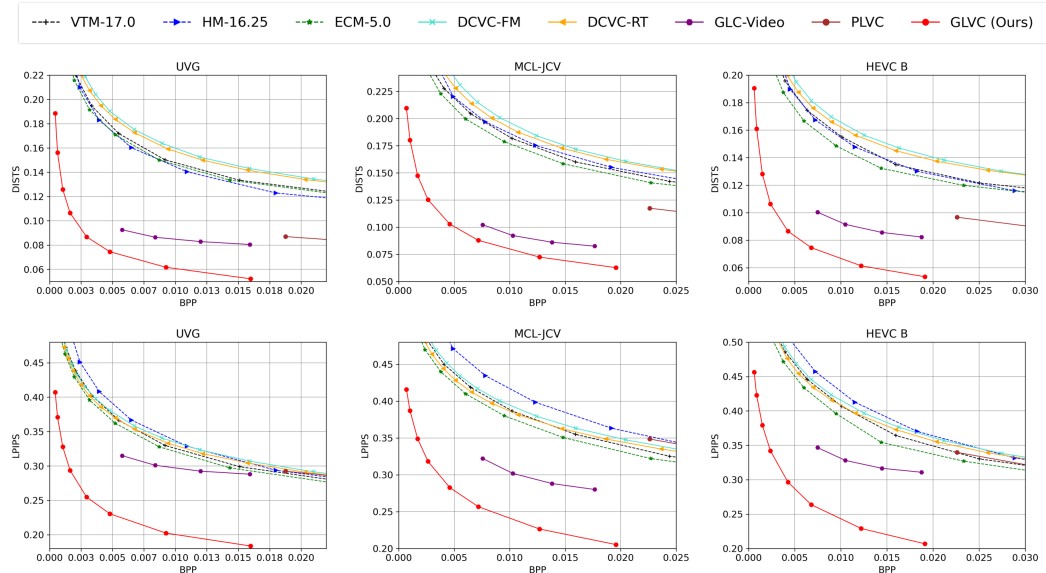

Figure 4: Rate-distortion curves in terms of DISTS and LPIPS.

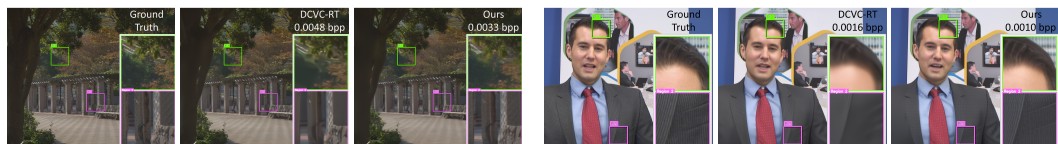

Figure 5: Visual comparisons of the decoded frames. Zoom in for best comparisons.

is known that these frame-level metrics are still not fully aligned with human perception. Therefore, next we present visual comparisons and conduct user study to evaluate the perceptual quality of different methods.

**Qualitative Comparisons**   Fig. 5 visualizes some decoded frames from different methods, showing that GLVC produces high-quality reconstructions even at extremely low bitrates. For instance, in one example (the right man in Fig. 5) where the entire 720p video is compressed to just 27 kbps (if compressed to 30 FPS), fine details like textures on clothing remain clear. However, previous method such as DCVC-RT got blurry reconstructions. Nevertheless, individual frame-wise qualitative comparisons still cannot reflect temporal coherence. Hence, we put a few decoded video clips in supplementary material and we recommend readers to play these videos, as it would be more clear that our approach is good at temporal coherence even with much lower bitrate.

**User Study**   We conducted a user study to compare the perceptual quality among our GLVC, the latest neural video codec DCVC-RT (Jia et al., 2025) and the traditional codec VTM. As shown in Fig. 6, our GLVC outperforms both other neural or traditional video codec at the similar rate. In particular, DCVC-RT with double rate could only win 47% against our GLVC, which means our method can sometimes compete with DCVC-RT at roughly half the rate.

## 5.3 DISCUSSIONS

**New Designs in Latent Video Codec**   The proposed GLVC incorporates several novel designs, some extending functionality, such as unified intra-/inter-frame compression and variable-rate control, and some improving compression performance. We conduct ablation studies to validate the performance contributions. As shown in Tab. 1, taking the full GLVC model as baseline and using DISTS as the evaluation metric, removing the recurrent memory mechanism increases bitrate by 9.2%. Tab. 2 further shows that, under extremely low rate, the common factorized hyperprior (Ballé et al., 2018) consumes a dominant share of the bits, whereas our parametric Gaussian hyperprior reduces this overhead. It pushes the lowest rate boundary of neural video codec.

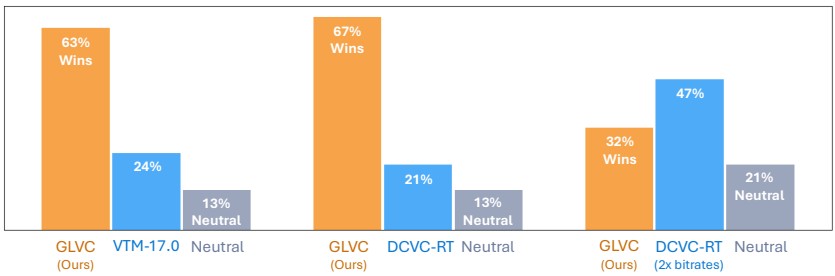

Figure 6: User study on UVG dataset. The proposed GLVC signicantly outperforms VTM and DCVC-RT at same bitrates, and is close to be competitive with DCVC-RT even at twice the bitrate.

| Model Variant | BD-rate |
|---|---|
| Baseline (Final Model) | 0 |
| w/o Memory Mechanism | + 9.2 % |

Table 1: Ablation on the proposed memoriza-tion mechanism. Positive values mean bitrate increase in terms of DISTS. Results are reported on Class B dataset.

| Settings | Rate of $y$ | Rate of $z$ |
|---|---|---|
| w/o GH | 0.00079 | 0.00128 |
| with GH | 0.00075 | 0.00072 |

Table 2: Ablation on parametric Gaussian hy-perprior. We calculate averaged rate cost (bit per pixel, bpp) of y and hyperprior z for all P-frames on Class B, where QP is very small.

**Effect of Finetuning**    After training the latent video codec, we finetune the reconstruction head of the latent codec, which only contains only 2.7M parameters. We provide RD curve comparisons in Appendix A.4. In our experiments, we also found that either skipping finetuning or over-finetuning (e.g., finetuning the tokenizer decoder as well) harms perceptual quality. We attribute this to the small finetuning scale, which may hurt the pretrained tokenizer's ability to synthesize perceptual details if over-optimized.

**Compression in Semantic Space vs. Pixel Space**    A natural question is whether our performance gains primarily stem from the latent compression framework or simply from GAN-based finetuning. To disentangle this, we applied the same finetuning strategy described in Sec. 3.3 to the previous state-of-the-art DCVC-RT (Jia et al., 2025). We found although finetuned DCVC-RT shows bet-ter results in metrics such as DISTS, it exhibits strong flickering artifacts visually, as shown in the decoded video shown in supplementary material. This contrast underscores the advantage of decou-pling semantic compression (appearance and motion) from perceptual detail synthesis.

**Continuous Tokenizer vs. VQ Tokenizer**    Beyond codec design, the choice of tokenizer is crit-ical. We found that even without latent compression, reconstructions from VQ tokenizers already exhibit strong temporal inconsistency, confirming that discretization disrupts smooth temporal dy-namics. Continuous tokenizers (such as SD-VAE (Rombach et al., 2022)) instead suppress flickering artifacts compared to VQ tokenizers. Continuous Wan video tokenizer further preserves the tempo-ral coherence. We provide video samples for comparisons in the supplementary material.

## 6    CONCLUSION

This paper introduces Generative Latent Video Compression (GLVC), a framework for percep-tually optimized neural video compression, which decouples perceptual detail synthesis from rate–distortion optimization by operating in a continuous semantic latent space. We redesign the codec for latent-domain compression and demonstrate the strong perceptual quality of the decoded results from GLVC. Much like latent diffusion transformed image generation, we hope GLVC opens a new path for perceptual video compression, which could be better aligned with human perception and modern media needs.

**Limitations**    Despite the strong performance of the proposed GLVC, so far GLVC cannot satisfy real-time compression for high-resolution videos. As shown in Appendix A.5, the current encod-ing/decoding complexity is still relatively high. In addition, due to the temporal downsampling in our used tokenizer, there is a four-frame latency for video streaming. However, these shortcomings could be remedied with future engineering, orthogonal to our contributions in this paper.

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

# A  APPENDIX

## A.1  HIERARCHICAL QUALITY ON LATENT VIDEO CODEC

Following (Li et al., 2024b), when training the latent video codec with the rate-distortion constraint, we apply hierarchical quality loss on different latent features to mitigate the temporal quality degradation. In our finally experiments, we set different weights for RD loss of different latents, which could be described as

$$\mathcal{L}_{\text{latent}} = \sum_{i=1}^{T} [\lambda \cdot \mathcal{R}(y_i) + \Delta(L_i, \hat{L}_i | y_i) * w_i], \tag{7}$$

where $w_1 = 8.0, w_2 = 1.2$ and later $w_i, i > 2$ repeated the value between $0.8$ and $0.5$. With the help of such a hierarchical quality loss, the temporal error propagation issue is largely mitigated, which is shown by the visualization of temporal quality across frames in Fig. 7. Note that as we use the pretrained Wan tokenizer (Team et al., 2025) that downsamples videos to 1/4 resolution in Temporal domain, we divide the rate of each inter latent feature by 4 as the rate cost per frame. Therefore, every 4 frames have the same rate cost in Fig. 7.

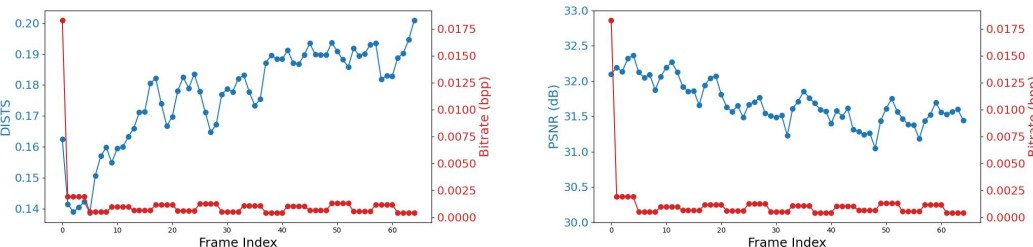

Figure 7: With hierarchical quality loss applied to train the latent video codec, our GLVC achieves relatively stable frame-wise quality in terms of DISTS or PSNR.

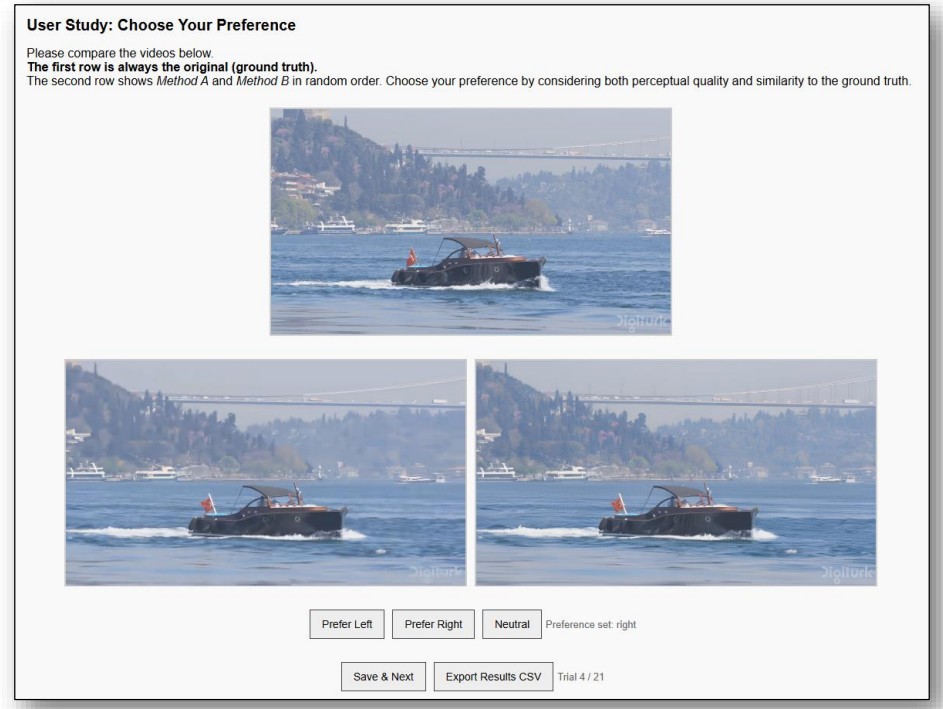

Figure 8: Screenshot of webpage used in the user study.

## A.2 USER STUDY DETAILS

To assess perceptual quality, we conducted a user study where raters compared videos side by side. In each trial, the original reference video was displayed at the top, while the bottom row showed two reconstructed videos: one produced by GLVC and the other by a baseline codec. The left/right order of methods was randomized to avoid bias.

Raters were instructed to select their preference based on both perceptual quality and similarity to the reference. Each trial thus yielded a direct pairwise preference between GLVC and the competing method. The only different setting with (Mentzer et al., 2022a) is that we provide an additional *neutral* option in case that users feel the competed methods have very similar quality. The aggregated preferences across participants were used to complement objective metrics, providing a more reliable measure of visual quality and temporal stability.

We invited more than 20 users and one third of them are women. Users are aged between 20 and 40. All users are paid fairly for this test.

## A.3 MORE RATE-DISTORTION CURVES AND BD RATE RESULTS

In Fig. 4, we present the rate-distorion curves on 1080p benchmarks (UVG, MCL-JCV and Class B). We further provide the results on low resolution benchamrks including HEVC Class C, D and E in Fig. 10. In terms of DISTS and LPIPS, the proposed GLVC significantly outperforms all previous model by a large margin, demonstrating its effectiveness across different video contents and resolutions.

For a more comprehensive comparison, we evaluate the proposed GLVC in terms of PSNR and MS-SSIM in Fig. 11 and Fig. 12. Compared to other perceptual video codecs like GLC-Video and PLVC, our GLVC achieves remarkable performance gain. Surprisingly, when compared to MSE optimized codecs at low bitrates, our GLVC presents competitive performance, especially on MS-SSIM. These demonstrate the superiority of GLVC in terms of semantic fidelity.

We provide the BD-Rate comparison in Tab. 4. Compared to VTM, our GLVC achieves an average bitrate saving of 91.5% and 89.4% in terms of DISTS and LPIPS, outperforming all compared codecs. In comparison, PLVC and GLC-Video achieves an average saving of 66.6% and 71.7% in DISTS.

## A.4 FINETUNING STRATEGY

The proposed GLVC applies a short finetuning procedure to adapt the reconstruction head of the latent video codec. Fig. 9 shows that the finetuning process boosts the performance in terms of both LPIPS and DISTS. Note that even if we do not apply the finetuning strategy, the GLVC (w/o finetuning) still could achieve strong results.

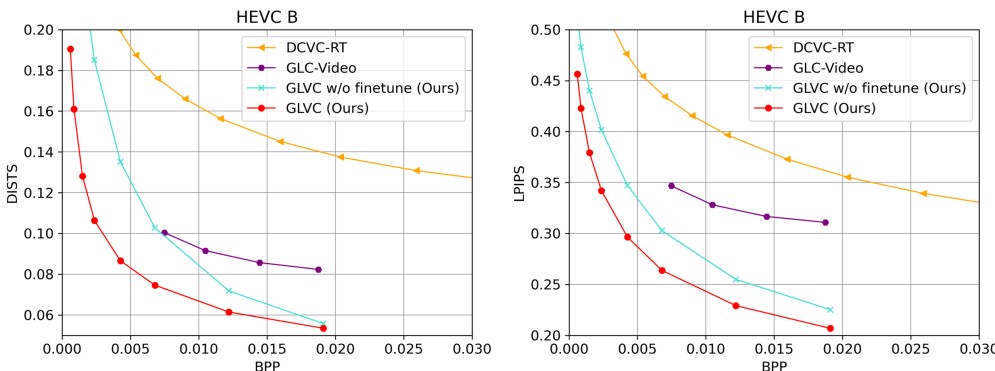

Figure 9: The finetuning strategy improves the performance in terms of metrics including DISTS and LPIPS.

## A.5 COMPLEXITY

In this section, we analyze the computational complexity of the proposed GLVC. The model contains 100.4M parameters. As shown in Tab. 3, GLVC achieves 194/306 ms per frame for encoding/decoding at $1920 \times 1080$ (5/3 fps) using a single A100 GPU. The latency decreases to 70/112 ms at $1080 \times 720$ (14/9 fps) and 29/45 ms at $640 \times 480$ (34/22 fps). These results demonstrate that GLVC delivers efficient and scalable performance across resolutions, satisfying real-time requirements at low video sizes.

Table 3: Coding speed per frames with different resolutions on an A100 GPU.

| Resolutions | $1920 \times 1080$ | $1080 \times 720$ | $640 \times 480$ |
|---|---|---|---|
| Encoding | 194 ms | 70 ms | 29 ms |
| Decoding | 306 ms | 112 ms | 45 ms |

## A.6 USE OF LLMs IN PAPER WRITING

In preparing this paper, Large Language Models (LLMs) were used solely for language polishing. They were not employed for tasks such as retrieval, research ideation, or other aspects of writing.

Table 4: BD-Rate (%) comparison in terms of DISTS and LPIPS.

| Method | BD-Rate on DISTS | | | BD-Rate on LPIPS | | |
|---|---|---|---|---|---|---|
| | UVG | MCL-JCV | HEVC B | UVG | MCL-JCV | HEVC B |
| *Traditional Codecs* | | | | | | |
| VTM-17.0 | 0.0 | 0.0 | 0.0 | 0.0 | 0.0 | 0.0 |
| HM-16.25 | -0.6 | 10.4 | 5.7 | 27.9 | 47.8 | 31.7 |
| ECM-5.0 | -9.3 | -13.1 | -18.8 | -12.5 | -17.0 | -17.8 |
| *MSE-Optimized NVCs* | | | | | | |
| DCVC-FM | 37.3 | 39.0 | 36.2 | 6.7 | 11.6 | 14.0 |
| DCVC-RT | 27.2 | 27.8 | 24.6 | 1.1 | -0.8 | 4.4 |
| *Perceptual-Optimized NVCs* | | | | | | |
| PLVC | -84.0 | -51.0 | -64.9 | -40.6 | -70.9 | -58.3 |
| GLC-Video | -42.1 | -86.1 | -86.8 | 11.9 | 5.3 | -4.2 |
| **GLVC (Ours)** | **-92.4** | **-91.1** | **-91.0** | **-88.8** | **-90.2** | **-89.3** |

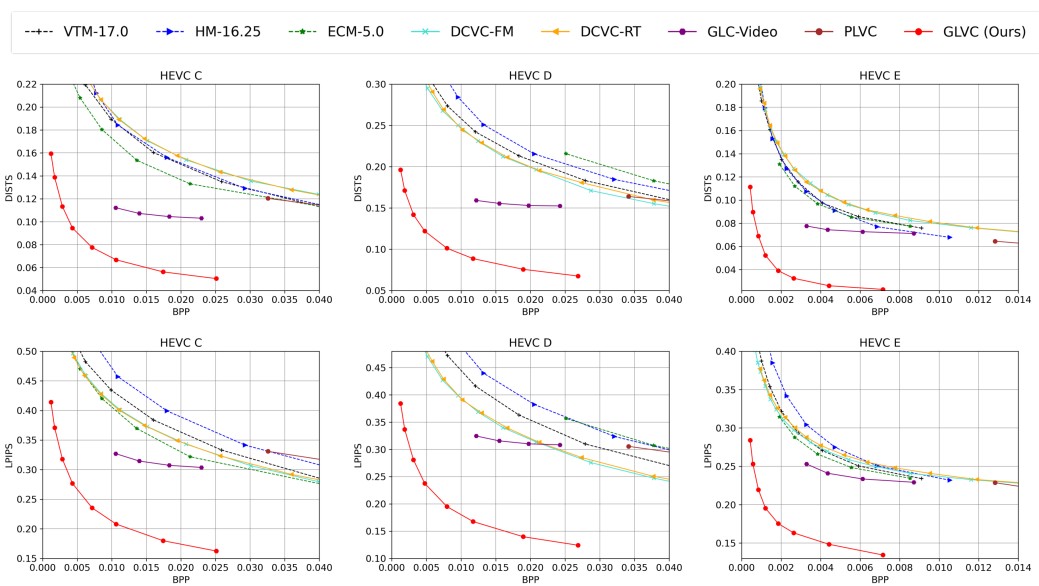

Figure 10: Rate-distortion curves in terms of DISTS and LPIPS on HEVC Class C, D and E.

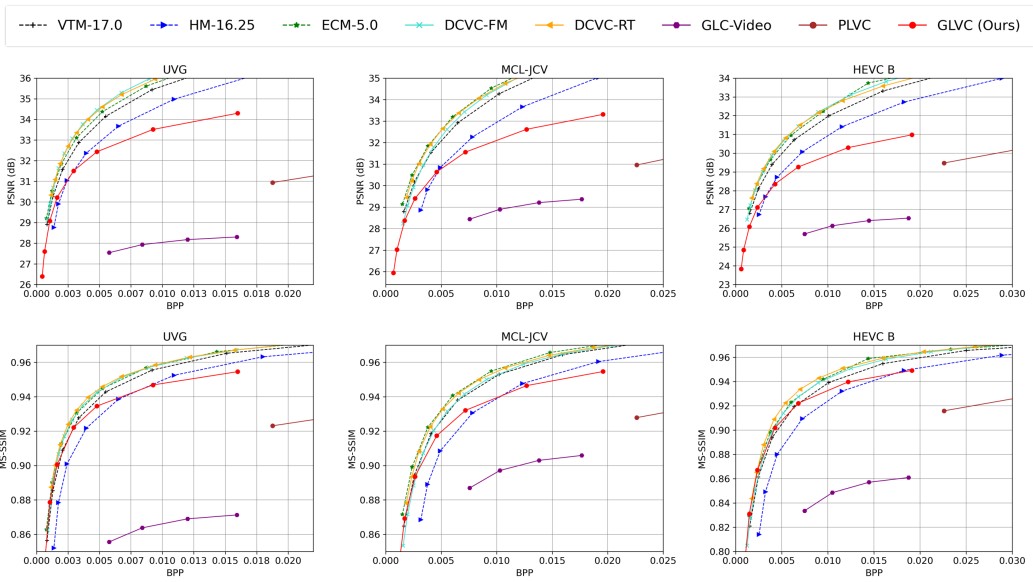

Figure 11: Rate-distortion curves in PSNR and MS-SSIM on UVG, MCL-JCV and HEVC Class B.

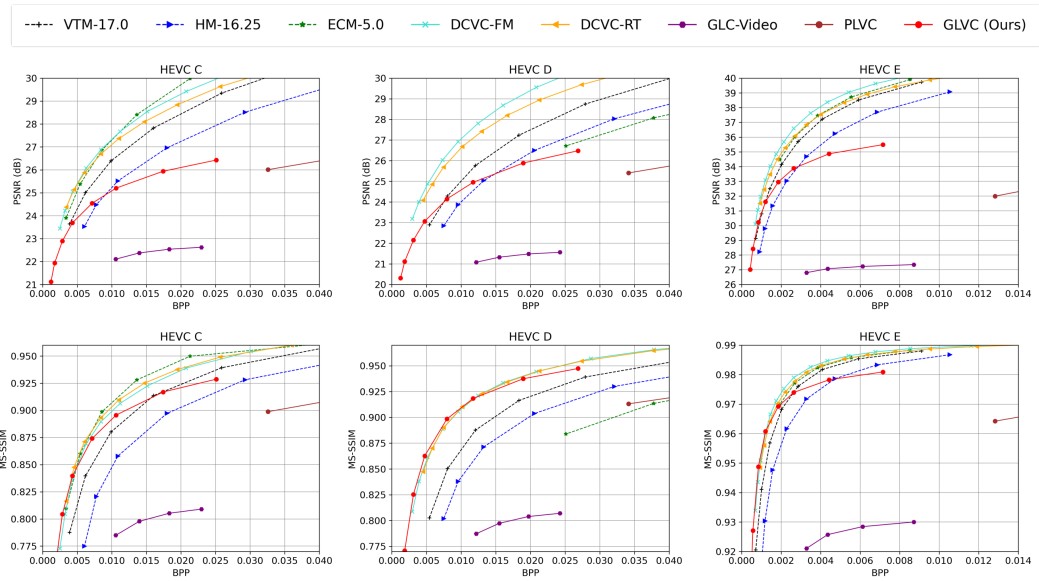

Figure 12: Rate-distortion curves in terms of PSNR and MS-SSIM on HEVC Class C, D and E.

