# OpenReview forum: "Generative Latent Video Compression"
_ICLR.cc/2026/Conference — Submitted to ICLR 2026_

### Official Review · Reviewer_kUyy · 2025-10-28

**Soundness:** 2
**Presentation:** 3
**Contribution:** 2
**Rating:** 4
**Confidence:** 5

**Summary:**

GLVC is a perceptual neural video codec which compresses the continuous video latents produced by a pre-trained Wan tokenizer and optimizes the rate-distortion trade-off in the latent space. It leverages several design choices, including unified/switchable intra-/inter-convolution-based conditional coding, recurrent memory, parametric Gaussian hyperprior, variable-rate quantization, and hierarchical quality. After RD optimization in the latent space, they Gan-finetune a small reconstruction head in the pixel space to further boost the perceptual quality. Results are reported on UVG, MCL-JCV, and HEVC classes with DISTS/LPIPS, and a small user study, both of which demonstrates the superiority of GLVC in retaining perceptual faithfulness.

**Strengths:**

- The paper is well-written and has a clear framing, where the insight of decoupling perceptual detail from rate-distortion-constrained optimization is interesting, and the switch from discrete VQ tokenizer to continuous alternatives is also different from previous previous image/video compression approaches.
- The experimental results are comprehensive and compelling: the BD-rate gains on DISTS/LPIPs across datasets are large; user study clearly favors GLVC over the benchmark VTM and DCVC-RT. Further, PSNR/MS-SSIM results are also included for completeness.
- The complexity profiling is transparent and detailed; the methods shows acceptable encoding and decoding runtime speed.

**Weaknesses:**

- **Novelty.** In my opinion, GLVC is built on a series of existing, well-established techniques. I am not sure about the benefits of unified intra- and inter-specific convolution layers design. Is it mainly for ensuring shared feature distribution and smoother transition between frame boundaries? How is this ablated (I don't think the current ablation by removing the memory mechanism does the job)? In terms of recurrent memory, there are plenty of NVC papers using 1D [1,2] or 2D [3,4] recurrent hidden states to ensure the spatiotemporal consistency and reduce long-term drifting (though they are not specifically tasked for perceptual purposes). The authors should more clearly demonstrate why the memory mechanism is different from the existing approaches or why it indeed reduces semantic drifting.
- **Metrics.** Following [5], I think the paper would benefit from more evaluation metrics, such as FID/FVD, measuring the consistency of semantic / spatial information using CLIP / IoU scores [6], or VMAF which is widely acknowledged in the video compression community and better balances distortion-perception trade-off.
- **Fairness.**: I think fine-tuning the reconstruction head with GAN is slightly asymmetric yet very impactful (based on Figure 9). The use of the custom 1M subset would certainly help, and the fact that applying the similar trick to DCVC-RT improves metrics but introduces flickering also indicate that GLVC benefits more cleanly from GAN finetuning given its tokenizer front-end. This should be more carefully discussed in the paper.
- **Conflicts in claims.** GLVC frames its training objective as rate-distortion optimization in a Gaussian latent space, but by replacing the likelihood with plain MSE to tackle the instability issue caused by tiny variance, the method breaks the claim as the codec no longer measures information loss in a meaningful "semantic" space; instead, it is just minimizing raw error in the tokenizer's feature domain?

Overall, I think the paper is of good quality and importance, especially considering there are relatively fewer perceptual-oriented video compression methods compared to the image compression topic. However, I am slightly concerned with its novelty considering much gains are yielded from foundational pre-trained models & fine-tuning on large datasets. I am willing to increase my score if the authors could discuss the insights & (potential) issues of adopting continuous instead of discrete VQ tokenizers.

[1] Learning for Video Compression with Hierarchical Quality and Recurrent Enhancement, CVPR'20

[2] HyTIP: Hybrid Temporal Information Propagation for Masked Conditional Residual Video Coding, ICCV'25

[3] ECVC: Exploiting Non-Local Correlations in Multiple Frames for Contextual Video Compression, CVPR'25

[4] GIViC: Generative Implicit Video Compression, ICCV'25

[5] Lossy Image Compression with Conditional Diffusion Models, NeurIPS'23

[6] Towards image compression with perfect realism at ultra-low bitrates, ICLR'24

**Questions:**

- It would be appreciated if the authors could provide additional details (e.g., visual examples, plots of flickering/forgetting/drifting) to better illustrate the advantage of continuous tokenization over discretized ones. Further, there are papers in video generation and compression (e.g., [4,7]) that performs generative modeling in the lossless pixel space, which might potentially be preferred, if 1) the mapping from pixel to latent space is still "lossy" 2) the continuous tokenizer's outputs don't have a good normal distribution (refer to **Weaknesses**)?

- Though I understand the perceptual codecs focus more on the (extreme) low rates, I am curious to knowing if the authors have conducted experiments for slightly higher bitrates, and when would the performance converge against the distortion-focusing baselines?

- How well does perceptual-oriented video codecs it preserves the semantic correctness and how would it impact machine vision tasks?

[7] High-fidelity image compression with score-based generative models.

---

> ### Author Response · Authors · 2025-11-13
> **Responses to Reviewer kUyy**
>
> We sincerely thank the reviewer for the thoughtful and detailed assessment. We address your concerns point-by-point below.
>
> 1. **Novelty**. We understand your concern about the novelty of this paper. GLVC indeed builds on a pretrained foundational tokenizer, and this is *intentional*. The core contribution of the paper is to demonstrate the benefits and implications of integrating a strong video foundation tokenizer into the rate–distortion framework. Much like latent diffusion reshaped image generation by operating in a semantically meaningful latent space, we show that latent video tokenizers can similarly transform video compression. Section 2 of the paper is dedicated to articulating this motivation and explaining why this paradigm shift is valuable for perceptual video coding.
>
> 2. **Continuous vs. discrete tokenizers**. We did examine this question both conceptually (Lines 201–209) and experimentally (Lines 466–471, with further visualizations in the Supplement). The evidence consistently shows that discrete VQ tokenizers cause temporal discontinuities, which leads to flickering and instability.
>
> 3. **Metrics**. Thank you for raising the metrics question. Classical image-level perceptual metrics such as FID, CLIP, or IoU do not reflect temporal flickering, which is central to perceptual video quality. VMAF, although widely used, behaves similarly to PSNR/MS-SSIM and focuses on frame-wise fidelity rather than perceptual coherence. FVD is indeed meaningful for temporal consistency, but it provides stable estimates only when a sufficiently large number of sequences are available for computing distribution-level statistics. For the common video compression benchmark datasets, there are too few sequences to get meaningful FVD results. Given these limitations, we chose to prioritize user study, which remains the most reliable way to assess perceptual fidelity in low-rate scenarios and allocates more experimental effort to that direction.
>
> 4. **Fairness of GAN finetuning**. Importantly, GLVC performs strongly even without any GAN refinement, which is quantitatively shown in Figure 9. The finetuning stage provides only a modest gain, mainly at extremely low bitrates where the MSE-based distortion in latent space can introduce noticeable artifacts. Its purpose is corrective rather than foundational, and GLVC’s advantage is not reliant on this step.
>
> 5. **Conflicts in claims**. As described in Lines 294–302, our original design employed maximum-likelihood–based distortion in the latent domain. However, in practice the very small variances of the latents led to numerical instability. To stabilize training, we replaced the likelihood with an equivalent MSE-form approximation, and subsequently applied a small pixel-space finetuning stage to resolve the mismatch between latent-domain error and pixel-domain semantics. This is a practical compromise rather than a conceptual departure from rate–distortion optimization.
>
> 6. **Performance at higher bitrates**. A limitation of using a pretrained video tokenizer is its inherent lossy transformation. Even without compression, tokenization–detokenization typically yields <35 dB quality (Fig. 11). But it is natural when we build perceptual codecs, we do not intend for near-lossless compression. While we acknowledge this limitation, perceptual video coding is explicitly targeted at low-rate, perceptually optimized regimes, where GLVC demonstrates clear advantages.
>
> 7. **Impact on machine vision tasks**. This is a great question. We believe GLVC offers unique advantages in the era of AIGC: videos generated by foundation models already originate in latent space, and GLVC can compress them directly without re-encoding from pixels. This enables future media pipelines—generation, editing, and compression—to operate coherently in a shared latent space with a common detokenizer, potentially improving both efficiency and cross-task consistency.
>
> We appreciate your comments that "our paper is overall of good quality and importance". Look forward to your further feedbacks.

---

### Official Review · Reviewer_AWDB · 2025-10-29

**Soundness:** 3
**Presentation:** 3
**Contribution:** 2
**Rating:** 4
**Confidence:** 5

**Summary:**

This work leverages a pretrained tokenizer to encode consecutive video frames into a latent space, enhancing perceptual and semantic information while addressing flickering issues in frame-wise models. Additionally, mechanisms such as the Recurrent Memory Mechanism are incorporated to capture long-term temporal information. The model achieves superior performance on perceptual metrics such as LPIPS and DISTS.

**Strengths:**

1.In the paper, motion is regarded as semantic information closely related to PSNR. By leveraging a continuous video tokenizer to compress multiple frames simultaneously, the approach alleviates issues with perceptual quality and inter-frame inconsistency.
2.The paper integrates the video tokenizer into the encoder through a series of improvements, balancing generation and reconstruction, and achieves significant performance gains on perceptual metrics.

**Weaknesses:**

1.Since multiple frames need to be processed simultaneously (4 frames during inference), there is an inherent delay
2.During training, 129 consecutive video frames are fed into the tokenizer, and the presence of the Recurrent Memory Mechanism further increases memory requirements.
3.Although the paper theoretically explains and demonstrates the method’s effect on temporal consistency, the evaluation of flickering artifacts remains largely qualitative. While the authors note that VQ tokenizers and perceptually optimized frame-wise models may induce flickering, the manuscript does not provide explicit visual examples or quantitative analyses to substantiate this effect. Including such evidence would make the impact on temporal consistency clearer and more convincing.

**Questions:**

1.The LPIPS metric in the paper is computed using a VGG network, which differs from the AlexNet commonly used in other tasks. Please clarify this choice in the manuscript.

---

### Official Review · Reviewer_bQxs · 2025-10-31

**Soundness:** 3
**Presentation:** 1
**Contribution:** 2
**Rating:** 2
**Confidence:** 5

**Summary:**

Neural video coding with perceptual quality optimization is challenging, even though neural codecs have advanced in recent years. Since neural video codecs aim to optimize perceptual quality directly, the authors propose to first use a pretrained tokenizer, which is optimized for perceptual quality, to map video frames from the pixel space to a continuous latent domain. Then, a compression model is used to code the latent directly. The proposed framework, GLVC, demonstrates improved perceptual quality compared with baseline methods.

**Strengths:**

- The proposed framework has a clear motivation, regarding the difficulty of optimizing the perceptual quality of learned video codecs.
- The proposed method demonstrates superior performance measured by perceptual metrics, including LPIPS and DISTS.
- A subjective experiment is conducted to validate the perceptual quality improvement of GLVC over the baselines.

**Weaknesses:**

- From an architecture perspective, employing the compression model in the latent domain is not a fundamentally novel idea. This can be treated as partitioning the encoder $E$ and decoder $D$ into two parts, i.e., $E(\cdot) = E_{latent}(E_{image}(\cdot))$ and $D(\cdot) = D_{image}(D_{latent}(\cdot))$. The only difference is the use of a pretrained encoder/decoder from a generative model.
- Recurrent mechanism for learned video codecs is not new. In fact, DCVC models are implicitly recurrent neural networks; the encoder and decoder take the previous state as input and output the new hidden state, which can implicitly learn long-range dependencies. Therefore, the claim (line 251) that "the temporal context is derived from only the previous frame" is not accurate.
- The proposed method introduces additional latency due to the tokenizer, which breaks the low-delay coding setting of the DCVC models that it is based on.
- Although the paper focuses on perceptual quality improvement, no analysis of the visual results is provided. For example, the presented results still contain some artifacts and temporal consistency issues, which are not discussed in the paper.
- Only a few samples are provided in the submission. In addition, only VTM and MSE-optimized model samples are included.
- The comparison to the learned model baselines (DCVCs) is not fair:
    - GLVC is trained on a larger-scale dataset, with up to 129 frames per sample.
    - GLVC utilizes a tokenizer, which increases both the complexity and the parameter count compared with the baseline.
- Technical details for the model are not provided. Including the network and layer design, hyper-parameters, etc.
- Ablation study is insufficient. Only one component is removed, which causes a 9% BD-rate increase, but this alone does not explain the superior performance of the proposed model.

**Questions:**

- What is the total computational complexity and model size (with and without the tokenizer)?
- Why is performance reported only for low-bitrate cases (e.g., < 0.02 bpp for the UVG dataset)? How does the proposed method perform at higher bitrates?
- Could you provide the detailed setting for removing the recurrent memory mechanism? What exactly does the model look like after removing this mechanism?
- Does the reported latency include the tokenizer? That is, are you measuring the full video compression time, or only the latent compression time?
- In the DCVC-RT paper, the authors found that the hyperprior contributes a large portion of the total bitrate (a similar finding to this paper) and thus introduced module-bank mechanisms to address this issue. Since GLVC has an architecture very similar to DCVC-RT, why not use the same idea for compressing the latent $z$? How does the proposed parametric Gaussian hyperprior compare with the one in DCVC-RT?

---

### Official Review · Reviewer_fwEH · 2025-11-05

**Soundness:** 2
**Presentation:** 3
**Contribution:** 2
**Rating:** 4
**Confidence:** 4

**Summary:**

This paper proposes Generative Latent Video Compression (GLVC), a neural video compression framework that separates semantic fidelity (rate–distortion optimization) from perceptual realism (handled by a pretrained generative tokenizer). The system first encodes frames into a continuous latent space using a pretrained video tokenizer (Wan), then applies a learned latent-domain codec and proposes two stage training. The main aim of the authors is to overcome the flicker artifacts in the perceptual compression, and in this paper the authors claim that operating in a continuous latent space mitigates the flicker artifacts commonly seen in perceptually optimized GAN-based video codecs.
Experiments report ~95% BD-rate savings over DCVC-RT on the perceptual metrics.

**Strengths:**

1) The idea of decoupling of semantic compression (appearance and motion) from perceptual detail synthesis is a clean, well-articulated idea. It aligns intuitively with the rate–distortion–perception (RDP) theory and generative latent modeling trends.

2) GLVC demonstrates substantial perceptual quality gains (LPIPS, DISTS) across several benchmarks and includes a user study that generally prefers GLVC over existing codecs. A good comparison with the traditional and neural codecs

3) The unified intra/inter codec with switchable layers and recurrent memory mechanism for capturing long term dependency. These contribute to temporal consistency and efficiency.

4) The focus on perceptual stability, the emphasis on reducing flicker and temporal inconsistency is timely and addresses a persistent issue in learned perceptual video compression.

**Weaknesses:**

Following are my major concerns:

1) My first concern is the novelty concern. There are works in the literature which perform compression on the continuous latent space with the pre-trained encoder. A highly relevant prior [1], already proposed compressing videos in a GAN’s latent space, performing inter/intra compression on continuous latent vectors and leveraging a pretrained generator for perceptual realism. Both offload perceptual synthesis to a pretrained generative model (GAN/tokenizer), and both perform rate–distortion optimization directly in the latent domain.
The authors main novelty is substituting the GAN latent (StyleGAN) with a pretrained continuous video tokenizer (Wan) and introducing a memory-augmented codec. Thus, this is an extension of the same paradigm with different generative model, and thus conceptual separation of  semantics vs. perception was already implicit in [1]. Despite being the relevant, the authors do not cite or discuss this paper  [1] in their related works.

2. The author states that latent space of the pretrained encoder (Wan tokenizer) removes the flickering artifacts in the perceptual compression, and also states that flickering is removed due to the memory adapter. If the flickering artifact is removed by the memory adapter, then the paper has some advantage. If the flickering is removed solely by the tokenizer, I believe that the paper does not propose additional knowledge compared to the literature.

3. perceptual metrics (like DISTS and LPIPS) used to study are some what limited and insensitive to the temporal artifacts because it measures only at the frame level.  No temporal perceptual metrics (e.g., FVD, t-LPIPS, CLIP-FVD) are reported.

4. The paper acknowledges that GLVC runs at ~5 fps encode / 3 fps decode (1080p) on an A100 GPU and adds a 4-frame latency due to temporal down sampling in the tokenizer. This is far from real-time and limits applicability. The authors minimize this issue as “orthogonal,” but in compression research, computational cost and latency are integral to the contribution.

5. Incomplete positioning in the literature. Beyond the missing GAN latent compression work, the paper omits discussions of:
Diffusion-based lossy codecs  [3], which also decouple perceptual realism from rate–distortion optimization. PLVC [2] and other GAN-based perceptual codecs’ mechanisms for temporal consistency.

6. For the high bitrate, the paper deviates very much from the fidelity according to the RD plots. The perceptual quality is important, but the fidelity is very important to preserve in the compression standards.

[1] “Video Coding Using Learned Latent GAN Compression” (ACM Multimedia, 2022),
[2] Perceptual Learned Video Compression with Recurrent Conditional GAN, https://arxiv.org/pdf/2109.03082v3
[3] Lossy Compression with Gaussian Diffusion, https://arxiv.org/abs/2206.08889

**Questions:**

1) The core problem of this paper is the removal or suppression of flickering artifacts, it is not clear from the paper whether it is removed by the pre-trained encoder or by the memory adapter ? Is it possible to quantify the portion of the flickering artifact removal by the memory adapter? Whether using this memory adapter to the existing method like DCVC -RT could remove the flickering artifacts.

2) The details of the memory adapter is not sufficiently described in the paper.

3) How the traditional codecs are compared in the paper. Traditional codecs are optimized for the YUV color space,  whether the output of the traditional codec is converted to the RGB space.

---

### Meta-Review · Area_Chair_fkiW · 2025-12-12

**Summary:**

Unanimity of the reviewers on the negative side. The authors provided responses to a number of concerns.
The ACs carefully read the paper, the reviews, and the authors' responses.
The paper requires major revision and a second review round.

**Reviewer Concerns:**

Unanimity of the reviewers on the negative side. The authors provided responses to a number of concerns.
The ACs carefully read the paper, the reviews, and the authors' responses.
The paper requires major revision and a second review round.

**Reviewer Scores:**

Unanimity of the reviewers on the negative side. The authors provided responses to a number of concerns.
The ACs carefully read the paper, the reviews, and the authors' responses.
The paper requires major revision and a second review round.

---

### Decision · Program_Chairs · 2026-01-26

Reject